# Primary-like Human Hepatocytes Genetically Engineered to Obtain Proliferation Competence as a Capable Application for Energy Metabolism Experiments in In Vitro Oncologic Liver Models

**DOI:** 10.3390/biology11081195

**Published:** 2022-08-09

**Authors:** Andrea Scheffschick, Jonas Babel, Sebastian Sperling, Julia Nerusch, Natalie Herzog, Daniel Seehofer, Georg Damm

**Affiliations:** 1Department of Hepatobiliary Surgery and Visceral Transplantation, University Hospital, Leipzig University, 04103 Leipzig, Germany; 2Saxonian Incubator for Clinical Translation (SIKT), Leipzig University, 04103 Leipzig, Germany; 3Department of General, Visceral and Transplantation Surgery, Charité University Medicine Berlin, 13353 Berlin, Germany; 4Faculty of Science, Brandenburg University of Technology Cottbus-Senftenberg, 01968 Senftenberg, Germany

**Keywords:** upcyte hepatocytes, primary human hepatocytes, HepG2 cells, HCC, energy metabolism

## Abstract

**Simple Summary:**

Fatty liver disease is an increasing health concern in Westernized countries. A fatty liver can lead to hepatocellular carcinoma (HCC), a type of liver cancer arising from hepatocytes, the major cells of the liver. How HCC may develop from the fatty liver is not known, and good cellular systems to investigate this are lacking. Recently, hepatocytes that can multiply continuously have been generated and suggested for hepatocyte research. In this study, we compared these continuously multiplying human hepatocytes to normal human hepatocytes and liver cancer cells, both within the state of fatty liver or not. We identified that these multiplying hepatocytes displayed many similarities to the liver cancer cells in terms of energy metabolism and concluded that these hepatocytes could be a pre-cancer model for liver cancer research and would be a valuable tool for HCC research.

**Abstract:**

Non-alcoholic fatty liver disease (NAFLD), characterized by lipid accumulation in the liver, is the most common cause of liver diseases in Western countries. NAFLD is a major risk factor for developing hepatocellular carcinoma (HCC); however, in vitro evaluation of hepatic cancerogenesis fails due to a lack of liver models displaying a proliferation of hepatocytes. Originally designed to overcome primary human hepatocyte (PHH) shortages, upcyte hepatocytes were engineered to obtain continuous proliferation and, therefore, could be a suitable tool for HCC research. We generated upcyte hepatocytes, termed HepaFH3 cells, and compared their metabolic characteristics to HepG2 hepatoma cells and PHHs isolated from resected livers. For displaying NAFLD-related HCCs, we induced steatosis in all liver models. Lipid accumulation, lipotoxicity and energy metabolism were characterized using biochemical assays and Western blot analysis. We showed that proliferating HepaFH3 cells resemble HepG2, both showing a higher glucose uptake rate, lactate levels and metabolic rate compared to PHHs. Confluent HepaFH3 cells displayed some similarities to PHHs, including higher levels of the transaminases AST and ALT compared to proliferating HepaFH3 cells. We recommend proliferating HepaFH3 cells as a pre-malignant cellular model for HCC research, while confluent HepaFH3 cells could serve as PHH surrogates for energy metabolism studies.

## 1. Introduction

Non-alcoholic fatty liver disease (NAFLD), as the hepatic manifestation of metabolic syndrome, is the most common cause of liver disease in Western countries, affecting up to one-third of the population [1,2]. Hepatic fat accumulation, termed liver steatosis, is the first stage of NAFLD, which can progress to non-alcoholic steatohepatitis (NASH) and further on to fibrosis and cirrhosis [3,4]. NAFLD is a major risk factor for developing hepatocellular carcinoma (HCC), which is considered the fourth most common cause of cancer-associated death worldwide. Surprisingly, approximately one-third of NAFLD-related HCC arises from a non-cirrhotic background, which complicates HCC diagnosis and treatment [5,6]. Due to a strong rise in obesity worldwide and a potential concomitant rise in HCC, there is a strong demand to understand NAFLD-related HCC pathomechanisms in order to develop potential treatments.

The molecular understanding of cancer is the subject of current research, and recently, changes in energy metabolism have been recognized as one of the core hallmarks of cancer development [7,8]. Due to their higher energy requirements because of enhanced proliferation and biosynthesis, cancer cells acquire changes in glucose metabolism to ensure high energy demands. The glucose transporter GLUT1 or enzymes of the glycolytic pathway like hexokinase 2 are frequently upregulated, enabling a higher glycolytic rate and hence, increased proliferation [9,10]. Rapidly proliferating cancer cells are usually characterized by the Warburg effect, which describes a higher non-oxidative glucose metabolism, even under aerobic conditions, resulting in higher lactate levels [11]. It was suggested that the Warburg effect might not be a characteristic solely for proliferating cancer cells but for strongly proliferating cells in general [12]. Regarding HCC, it is not yet known whether the proliferation of hepatoma cells in HCC can be discriminated from the proliferation of regenerating hepatocytes.

For studying HCC pathomechanisms, the liver cancer cell lines HepG2 and Huh7 are the standards of in vitro liver cancer studies [13,14]. HepG2 cells are the most extensively used cellular system and have been used in NAFLD in vitro models [15,16]. Furthermore, HepG2 cells demonstrate important characteristics of cancer cells, including loss of contact inhibition, aberrant glucose metabolism and an altered insulin signaling pathway machinery [15,17,18]. The use of primary human hepatocytes (PHHs) is the gold standard in liver research [19]; however, PHHs do not proliferate and are therefore not suitable as a cancer cellular model [20,21]. Advances have been made with respect to hepatocyte immortalization in order to gain a cellular system with continuous proliferation [22]. Immortalized hepatic cell lines have been generated from adult or fetal human hepatocytes by the transduction or transfection of oncogenes or telomerase reverse transcriptase (TERT) [23,24]. As a new promising alternative in addition to the previously mentioned cellular models, genetically modified PHHs using upcyte technology have recently been introduced into research. Human hepatocytes are genetically modified by proliferation-inducing genes introduced via lentiviral transduction, resulting in the so-called upcyte hepatocyte cells [25]. These cell lines have been validated as feasible alternatives in in vitro liver cell systems [26]. However, due to the strong increase in proliferation in HepaFH3 cells, it is still not known whether the HepaFH3 cells are an appropriate replacement for PHHs or a better replacement for oncological cellular systems in terms of energy and metabolic characteristics.

With this study, we sought to compare the HepaFH3 cells with continuous proliferation to PHHs and HepG2 cells in terms of energy and metabolic characteristics. In addition to proliferating HepaFH3 cells, we analyzed HepaFH3 cells in the confluent state to analyze the metabolic capacity of HepaFH3 cells in a state where energy metabolism is not exhausted due to excessive proliferation. Furthermore, we investigated all cell types within the disease state of steatosis using an in vitro model system to determine whether HepaFH3 cells are suitable surrogates for PHHs or HepG2 cells for studying NAFLD pathomechanisms in an in vitro steatosis cellular model. Collectively, we show with our study that proliferating HepaFH3 cells show similarities to HepG2 cells in terms of energy metabolism, suggesting proliferating HepaFH3 cells as a pre-malignant cellular system for HCC research.

## 2. Materials and Methods

### 2.1. Chemicals and Material

William’s E Medium with GlutaMAX, supplements (FCS (fetal calf serum), penicillin/streptomycin, HEPES, MEM NEAA, Na-pyruvate, L-glutamine) and PBS (Mg^2+^/Ca^2+^ +/+ and −/−) were purchased from Gibco (Paisley, UK). DMEM media was obtained from GE Healthcare (Pasching, Austria). Human insulin was purchased from Sanofi Aventis (Frankfurt am Main, Germany) and dexamethasone from Merck (Darmstadt, Germany). Trypsin was obtained from PAN-Biotech. Percoll and Trypan Blue were provided by Biochrom (Berlin, Germany). If not stated differently, all other chemicals were purchased from Sigma Aldrich (Munich, Germany). Rat tail collagen was prepared in our laboratory according to the protocol established by Rajan et al., *Nat. Protoc.* 2006 [27]. HepG2 cells were obtained from Leibniz Institute DSMZ—German Collection of Microorganisms and Cell Cultures. The PHH and HepaFH3 culture medium was based on William’s E Medium with GlutaMAX, supplemented with 10% FCS, 32 U/L insulin, 15 mM HEPES, 0.1 mM MEM NEAA, 1 mM Na-pyruvate and 1 µg/mL dexamethasone. HepG2 culture medium was based on DMEM with 10% FCS and 2 mM L-glutamine. PHH and HepaFH3 control medium, which was used for steatosis induction in PHHs and HepaFH3 cells, was based on William’s E Medium E with GlutaMAX with 5% FCS, 15 mM HEPES, 0.1 mM MEM NEAA, and 1 mM Na-pyruvate. HepG2 control medium, which was used for steatosis induction in HepG2 cells, was based on DMEM with 2% FCS, 1% BSA, and 2 mM L-glutamine. All culture media were supplemented with 100 U/100 µM penicillin/streptomycin before use.

### 2.2. Isolation of Primary Human Hepatocytes

For hepatocyte isolation, liver tissues were obtained after informed consent was received from patients undergoing liver surgery at the Department of General, Visceral and Transplantation Surgery at Charité University Medicine Berlin, Germany and the Clinic and Polyclinic for Visceral, Transplantation, Thoracic and Vascular Surgery at University Clinic Leipzig. The study was conducted according to the Declaration of Helsinki and received prior approval from the local ethics committee (Charité University Medicine Berlin, registration number EA2/076/09, date 28 July 2009). Tissue samples were freshly retrieved from macroscopically tumor-free areas of the resected livers (Table 1). PHHs were isolated from liver tissue samples as described previously [28,29]. In brief, PHHs were isolated from liver tissue by a two-step EGTA/collagenase perfusion technique. PHH were pooled and washed using PBS (Mg^2+^/Ca^2+^ +/+) and finally resuspended in PHH culture medium William´s E medium with GlutaMAX supplemented with 10% FCS, 15 mM HEPES, 1% MEM NEAA, 1 mM Na-pyruvate, 1 µg/mL dexamethasone, 32 U/L insulin and 100 U/100 µM penicillin/streptomycin. PHH were directly seeded on collagen-coated plates and cultivated or snap-frozen in FCS +10% DMSO in liquid nitrogen. Thawing of cryopreserved PHHs was accomplished by washing the cells in 50 mL complete culture medium followed by centrifugation at 51 g and resuspension in an appropriate amount of culture medium.

### 2.3. Generation of Upcyte Hepatocytes

The donor from whom PHHs for transduction into HepaFH3 were obtained was a 59-year-old woman with liver metastasis from metastatic cancer of unknown primary (CUP), obesity class I. No secondary diagnoses were known; she had undergone no chemotherapy and demonstrated serum without pathological findings and no prolonged medication. Transfection and expansion were performed by the workgroup of Jan-Heiner Küpper [26]. In brief, PHH with a viability of 85% were seeded on collagen-coated culture plates with a density of 3 × 10^4^ cells/cm^2^. Hepatocytes were genetically modified using Medicyte’s proprietary upcyte technology [30]. After 24 h of attachment, cells were transduced with viral particles containing the proliferation-inducing genes. The cells were cultured for additional 3 to 8 weeks. During this period, the used upcyte Hepatocyte Growth Medium was replaced every 2–3 days. After 10 days, the first proliferating colonies were observed in the upcyte treated cell cultures. Once upcyte hepatocytes were clearly evident in the cultures, they were trypsinized and reseeded at a density of 2 × 10^4^ cells/cm^2^ and allowed to expand up to 80% confluence. Subsequent passages were performed using a seeding density of 5 × 10^3^ cells/cm^2^ [25]. Cells were cultivated in HepaFH3 culture medium.

### 2.4. In Vitro Cell Culture and Induction of Steatosis

PHHs and cell lines were cultured at 37 °C, 5% CO_2_ in a humidified incubator. To induce steatosis, respective cells were treated for 24 h with a 1 mM mixture of oleic and palmitic acid in a ratio of 2:1 [31]. In order to detect lipids, cultured cells were stained with Oil Red O and normalized to sulforhodamine B (SRB) protein staining.

For the different cell types, different cell numbers were seeded for experiments because, despite their hepatic origin, all our used cells have different sizes and, consequently, different protein content. Hence, the different cell types were seeded so as to reach a confluency of around 80–100%. We choose this small window of confluency as a compromise between the needed functionality and comparable cell numbers. PHHs were seeded to about 100% confluency, which was reached by adding a surplus of 200,000 cells/cm^2^ to the cell culture vessel. Proliferating HepaFH3 cells and HepG2 cells were cultured at approximately 80% confluency, for which 250,000 cells/cm^2^ for proliferating HepaFH3 cells and 100,000 cells/cm^2^ of HepG2 cells were seeded. For confluent HepaFH3 cells, 250,000 cells/cm^2^ of HepaFH3 were seeded, and upon 100% confluency, the cells were cultured for a further seven days until experiments were performed.

### 2.5. Oil Red O Staining

In order to detect neutral lipids in all cultivated cells, cells were washed with PBS and fixed with a 4% formaldehyde solution (Herbeta, Berlin, Germany) for at least 5 min. Afterwards, they were covered with Oil Red O in 0.2% concentration in a 2:3 isopropanol/H_2_O mixture for 15 min. Unbound dye was removed by washing with water. After air drying, isopropanol was used to resolve the stain. Absorbance was measured at λ = 500 nm using a microplate photometer (FLUOstar OPTIMA, BMG Labtech, Ortenberg, Germany) and normalized by sulforhodamine B protein staining.

### 2.6. Sulforhodamine B Protein Staining

For normalization of Oil Red O staining, surface protein content had to be determined. Cells were covered with sulforhodamine B at a final concentration of 0.4% in a 1% acetic acid solution incubated in the dark for 30 min at RT. Cells were washed four times with a 1% acetic acid solution. For quantification, SRB was resolved with 10 mM unbuffered TRIS solution, and absorbance was measured at λ = 565 nm using a microplate photometer.

### 2.7. Western Blot Analysis

Cells were lysed in RIPA-buffer (based on PBS and containing 50 mM Tris-HCl, 0.1% sodium dodelcyl sulfate (SDS), 0.5% Triton X100, protease inhibitor complete (Roche), 0.5 M NaF, 100 mM Na_3_VO_4_, and 1 M β-glycerol phosphate at pH 7.5) and frozen at −80 °C. After thawing, cell debris was removed by centrifugation at 9000× *g* for 10 min, and protein determination was performed using a bicinchoninic acid (BCA) protein assay kit (Thermo Scientific) as described below. Sample concentration was adapted to 1 µg protein/µL with dH_2_O and 5× loading buffer containing 0.4 M Tris base, 10% SDS, 25% β-mercaptoethanol, 50% glycerol and boiled for 5 min at 96 °C. Samples were run on 10% acrylamide gels, and proteins were transferred onto PVDF membranes (Amersham, Buckinghamshire, UK) and blocked with 5% BSA in TBS-Tween (0.1%) for 2 h. The following primary and secondary antibodies have been used: 5 hosphor-P44/42 MAPK (ERK1/2) (Thr202/Tyr204) (clone 197G2, Cell Signaling Technologies, Danvers, MA, USA), 5 hosphor-AKT (Ser473) (clone DE9, Cell Signaling Technologies), 5 hosphor-GSK3 (Ser21/9) (clone D17D2, Cell Signaling Technologies), 5 hosphor-Foxo1 (Ser256) (Cell Signaling Technologies), β-Actin (clone D6A8, Cell Signaling Technologies), anti-rabbit-HRP (Cell Signaling Technologies) and anti-mouse-HRP (Jackson Immuno Research, PA, US). Primary antibodies were diluted in 5% BSA in TBS-Tween (0.1%) and incubated at 4 °C overnight. Following washing steps with TBS-Tween (0.1%), secondary antibodies were incubated for 1 h at RT; for visualization, Pierce ECL substrate (Thermo Scientific) and the imager Chemi Doc MP System (Bio-Rad) were used. Quantification of protein bands was carried out using the Image Lab software (Bio-Rad). In previous studies, it was shown that steatosis does not profoundly affect the total protein level of, e.g., ERK, Foxo1 or Akt, which is why we did not normalize the phospho-protein levels to the total protein levels but to the housekeeping protein β-actin [32,33]. All Western blots for pFoxo1 and beta-actin were taken with the same exposure time. For pAkt, pERK and pGSKα, β different exposure times were used for different blots, but this was corrected for in the evaluation of the band intensity.

### 2.8. Quantification of Total Protein

For measurement of total protein content, a BCA assay from Thermo Scientific was performed. A total of 20 µL of the sample was mixed with 300 µL 4% copper sulfate solution with BCA-Reagent (1:50) (Thermo Scientific, Waltham, MA, USA) and incubated for 30 min at 37 °C. Absorbance was measured at λ = 565 nm by a microplate photometer. Bovine serum albumin standard with a concentration of 2 mg/mL was used to establish the standard curve.

### 2.9. Cellular Metabolic Activity

The MTT assay was used as a surrogate parameter to estimate cellular metabolic activity [34]. Cells were washed twice with PBS and covered with sterile filtered MTT working solution (0.005% Thiazolylblue in PBS and FCS in a 1:10 ratio) for 2 h at 37 °C and 5% CO_2_. The supernatant was carefully removed, and 100 µL solubilization solution (5 g SDS, 49.7 mL DMSO, 0.3 mL acetic acid) was added. Absorbance was measured at λ = 570 nm using a microplate photometer.

### 2.10. Glucose

For glucose determination, a glucose assay from Analyticon (Fluitest GLU Glucose, cat. no. 4341) was used according to the manufacturer’s instructions. In brief, 5 µL of sample was incubated with 200 µL volume R1 solution for 15 min at 37 °C. Absorbance was measured at λ = 550 nm by a microplate photometer.

### 2.11. Glycogen

For glycogen determination, an enzymatic essay was used. Cells were washed twice with PBS and detached from plates using trypsin/EDTA solution (Biochrom). Cell detachment was stopped by PBS with 20% FCS followed by centrifugation at 300× *g* for 5 min at 4 °C. Resuspended cells were sonicated for 10 min. A total of 10 µL of sample was mixed with 3.6 µL of 7% perchloric acid and 20 µL of 0.5 M sodium hydroxide, and the suspension was incubated at 100 °C for 10 min, followed by centrifugation at 500× *g* for 1 min at 4 °C. Next, 20 µL of 2 M acetate buffer at pH 4.5, containing 4 mg/mL amyloglucosidase (70 U/mg from Sigma), was added to the solution and incubated for 2 h at 55 °C. Then, the glucose assay (Analyticon) was used to determine the glucose concentration.

### 2.12. Urea

For urea determination, an assay according to Zawada et al. was established [35]. The following reagents were used. *O-Phthalaldehyde reagent:* 80 mL dH_2_O were mixed carefully with 7.4 mL 2.5 mol/L H_2_SO_4_ on ice. 60 mg o-Phthalaldehyde and 100 µL 30% Brij-35 (Thermo Scientific) were added to the cooled solution and diluted with dH_2_O to 100 mL. *Primaquine reagent:* 500 mg boric acid was dissolved in 60 mL dH_2_O and mixed carefully with 22.2 mL 2.5 mol/L H_2_SO_4_ on ice. 102.6 mg Primaquine and 100 µL 30% Brij-35 were added to the cooled solution and diluted with dH_2_O to 100 mL. For the urea assay, 50 µL of sample was mixed with 100 µL of O-Phthalaldehyde and shaken gently. Afterwards, 100 µL of primaquine reagent was added and incubated for 1 h at 37 °C. Absorbance was measured at λ = 430 nm by using a microplate photometer.

### 2.13. ALT/AST

In order to determine the levels of enzymes alanine transaminase (ALT) and aspartate transaminase (AST), Fluitest Kits from Analyticon (ALT: 1186, AST: 1176) were used according to the manufacturer’s instructions. In brief, 20 µL of sample was mixed with 200 µL of reaction mixture (R1 and R2 solution in a ratio of 1:1). Bio-Cal E (Analyticon) was used for establishing the standard curve. Absorbance was measured at λ = 430 nm by a microplate photometer.

### 2.14. Albumin

An enzyme-linked immunosorbent assay (ELISA) was used to determine the albumin concentration in the cell culture medium. ELISA-compatible 96-well plates (Sarsted) were coated for 1 h at RT with affinity-purified goat anti-human albumin coating antibody (Bethyl Laboratories) at 1:100 in coating buffer with 50 mM KHCO_3_ (pH 9.6). Afterwards, wells were washed four times with a washing buffer containing 0.05% Tween 20, 140 mM NaCl and 50 mM Trizma base at pH 8.0. Blocking was performed using washing buffer without Tween 20 but with 1% of BSA (Sigma) for at least 30 min. For the standard curve, human reference serum (Bethyl Laboratories) was diluted with dilution buffer (equal to washing buffer, containing 1% BSA). Wells were incubated with 100 µL of sample or standard for 1 h at RT and afterwards washed with washing buffer four times. Horseradish peroxidase-conjugated goat anti-human albumin detection antibody (Bethyl Laboratories) was diluted at 1:75,000 in dilution buffer, and 100 µL was applied to wells for 1 h at RT. After incubation, the suspension was removed, and the wells were washed. TMB (3,3′,5,5′-tetramethylbenzidine) and one component HRP microwell substrate (Bethyl Laboratories) were added. The reaction was stopped after 15 min with 1% sulfuric acid. Absorbance was measured at λ = 450 nm by a microplate photometer, and OPTIMA software (BMG Labtech) was used to plot the standard curve using the 4-parameter fit and log axis.

### 2.15. Lactate Assay

Lactate levels were determined in the cell supernatants using an enzymatic assay from Dialab (D08130). The assay was performed according to the manufacturer´s instructions. Briefly, 2 µL of sample were mixed with 200 µL of reagent mix (4:1 ratio of R1 and R2). Following incubation for 5 min at 37 °C, the absorbance was measured at 340 nm at 37 °C using a plate reader.

### 2.16. Statistical Analysis

The datasets were analyzed with two-way ANOVA using Graph Pad Prism 8 software. All experiments were at least performed in triplicates, except for the analysis of albumin and lactate in HepG2 cells, which was only performed once. Results are given as mean ± SEM. Differences were considered significant at *p* < 0.05 calculated by Tukey´s or Sidak’s multiple comparisons test. ns = *p* ≥ 0.05, * = *p* < 0.05, ** = *p* < 0.01, *** = *p* < 0.001, **** = *p* < 0.0001.

## 3. Results

With this study, we aimed to investigate the application of upcyte hepatocytes using our HepaFH3 cell clone as a model of pre-malignant liver lesions. Therefore, we characterized the HepaFH3 cell line in two different cultivation methods—under normal and steatotic conditions—and compared major metabolic functions between differentiated liver cells (PHH) and proliferating hepatoma cell lines (HepG2).

### 3.1. HepaFH3 Show Rather Morphologic Similarities to HepG2 Cells Than to PHHs

For our studies, HepaFH3 cells were studied in two different states—in the state of proliferation and in the state of confluency when proliferation has ceased and differentiation can occur. In order to study cells within the disease state of steatosis, cells were treated for 24 h with oleic and palmitic acid (2:1 ratio) according to an established protocol [31]. Before steatosis induction, a few small lipid droplets were visible in PHHs, as seen by light microscopy (representative image of all control cells in Figure 1A,C,E,G).

In general, PHHs were bigger than HepaFH3 cells and HepG2 cells, and nuclei within PHHs and their partially polyploid state were visible. The cell morphology of HepaFH3 cells was denser, and hence, the cells were smaller as compared to PHHs and generally more similar to HepG2 cells. For confluent HepaFH3 cells, cells reached a high confluence, visible by the fact that cell borders are difficult to see. Upon steatosis induction, the number of small lipid droplets clearly increased in PHHs, which was less obvious visually for HepaFH3 cells and HepG2 cells due to their smaller cell size (representative image of steatotic cells in Figure 1B,D,F,H). Furthermore, some cell rounding and detachment in proliferating HepaFH3 cells were visible after steatosis induction.

### 3.2. PHHs, HepaFH3 and HepG2 Cells Exhibit the Same Ability of Lipid Storage

The lipid storing capacity was measured using Oil Red 
O staining normalized to protein content. Before steatosis induction, PHHs, 
proliferating HepaFH3, confluent HepaFH3 and HepG2 cells showed strongly varying lipid content levels in the initial (overnight after cell seeding) and control cells (further 24 h culture), with PHH cells having the highest initial lipid content (Figure 2A). 

Following steatosis induction, all cell types showed the same capacity to store lipids, with higher lipid content of about 3-fold in the steatotic state compared to the control state in all cell types. The overall lipid content still varied strongly between different cell types, and PHHs still had the highest lipid levels. Significant induction of lipotoxicity as measured by the release of LDH and the transaminases AST and ALT as a result of impaired membrane integrity was not observed. However, slightly higher levels of LDH upon steatosis induction released by PHHs and confluent HepaFH3 cells and for steatotic PHHs for AST could also be detected. Nevertheless, steatosis did not significantly induce cell toxicity (Figure 2B–D). Taken together, our liver models showed comparable capacities for lipid accumulation without manifestation of lipotoxic effects, indicating that these in vitro cellular systems can be used to study the effects of steatosis on metabolism.

### 3.3. HepaFH3 and HepG2 Cells Show Strongly Diminished Levels of Basic Hepatocyte Markers Compared to PHHs

In order to investigate whether the HepaFH3 cells retained basic hepatic functions, we analyzed the ability of the cells to produce the classical hepatic markers albumin, urea and the intracellular transaminases AST and ALT. As expected, PHHs exerted high levels of albumin, urea and the transaminases AST and ALT (Figure 3A–D), while proliferating HepaFH3, confluent HepaFH3 and HepG2 cells had strongly reduced levels of these markers.

HepaFH3 cells were almost completely devoid of albumin, suggesting a dedifferentiated state of this cell line. Regarding urea levels, HepaFH3 cells and HepG2 cells exhibited very low but detectable urea levels. With increasing passage number, albumin levels but not urea levels decreased in both confluent and proliferating HepaFH3 cells (Appendix A). Interestingly, AST and ALT levels were lowest in proliferating HepaFH3 cells and higher in confluent HepaFH3 cells, showing that confluent HepaFH3 cells exhibit closer similarities to PHH in terms of transaminase levels. A passage number-dependent effect on transaminase levels was not detected (Appendix A). Steatosis significantly upregulated AST and ALT in PHH cells, while other hepatic markers were not affected. Overall, HepaFH3 cells showed a closer relationship to HepG2 cells in terms of these basic hepatic markers than to PHH cells.

### 3.4. Proliferative Rate and Glucose Metabolism Are Elevated in Proliferating HepaFH3 and HepG2 Cells Compared to Confluent HepaFH3 and PHH

Next, we assessed the proliferative capacity of cells by measuring the total protein content, assuming that with proliferation, the overall protein mass increases. Both proliferating HepaFH3 and HepG2 cells showed an increased protein content over time when comparing initial cells (12 h after cell seeding) and control cells (further 24 h cell culture), suggesting proliferation within this time span (Figure 4A).

On the contrary, PHHs showed reduced protein content over time, suggesting cell loss. Interestingly, confluent HepaFH3 cells exhibited no increase in protein mass over time, which proposes the capacity of contact inhibition. Steatosis resulted in lower protein content in proliferating HepaFH3 cells, while for the other cell types, an effect of steatosis on protein content could not be observed. Regarding the metabolic activity as measured by MTT assay, HepG2 cells had the highest metabolic activity. The metabolic activity in proliferating and confluent HepaFH3 cells was comparable, and PHHs had the lowest metabolic activity (Figure 4B). Interestingly, steatosis resulted in a higher metabolic rate in HepaFH3 cells. Collectively, HepaFH3 cells in the proliferating state showed strong similarities to HepG2 cells in terms of proliferation, while confluent and proliferating HepaFH3 cells ranged in between HepG2 and PHHs regarding their metabolic activity.

Since HepaFH3 and HepG2 cells exerted a higher metabolic rate than PHH, we asked whether the carbohydrate metabolism as an energy source is affected within HepaFH3 cells. We assessed the ability of cells for glucose uptake from the culture medium, which equals their glycolytic activity (Figure 4C). Strikingly, HepaFH3 cells had a higher glucose uptake ability than PHHs, which was higher in proliferating HepaFH3 cells as compared to confluent HepaFH3. PHHs did not take up glucose from the medium but rather released glucose, potentially an indicator of gluconeogenesis. HepG2 cells had the highest glucose uptake capacity among control cells. Steatosis strongly induced glucose uptake in confluent HepaFH3 cells but not in other cell types. Regarding glucose storage capacity, HepG2 cells showed no signs of glucose storage in terms of glycogen production (Figure 4D). Glycogen was also almost undetectable in proliferating HepaFH3 cells but present in confluent HepaFH3. Still, glycogen storage levels in confluent HepaFH3 were lower as compared to PHH. In line with the high glucose uptake rate of proliferating HepaFH3 and HepG2 cells, these cells also showed higher lactate levels compared to PHHs (Figure 4E). Interestingly, HepaFH3 cells showed even higher lactate levels in the steatotic state. Conclusively, HepG2 and proliferating HepaFH3 cells seem to have a higher glycolytic rate, increased lactate levels and almost no glycogen storage compared to PHHs. Confluent HepaFH3 cells showed more similarities to PHH cells than to HepG2 cells in terms of glycogen storage.

### 3.5. Increased Phosphorylation of AKT Kinase, ERK and Foxo1 of Proliferating HepaFH3 and HepG2 Cells Suggests Increased Proliferation and Survival

Since increased proliferation and increased energy metabolism were detectable for proliferating HepaFH3 cells and HepG2 cells, we sought to analyze the phosphorylation and hence activation status of related key signaling cascade kinases using the Western blot technique (Figure 5 and Appendix A). As lipid-induced insulin resistance is a hallmark of fatty liver disease, we additionally analyzed the phosphorylation status of kinases following stimulation with insulin.

PI3 kinase/AKT kinase are critical for cell growth, survival as well as metabolism and are regarded as key kinases in cancer regulation [36]. Proliferating HepaFH3 and HepG2 cells already showed high AKT phosphorylation in the unstimulated state, which was higher compared to PHHs (Figure 5A), being in line with the potentially higher proliferation, higher glycolysis rate and lactate levels of these cell types (Figure 4A). However, insulin-induced AKT phosphorylation was observed only in proliferating HepaFH3 and PHHs, while confluent HepaFH3 and HepG2 were not responsive.

GSK3α,β is a downstream signaling kinase of the PI3 kinase/AKT kinases, which block GSK3α,β activation. GSK3α,β is essential in glycogen metabolism by phosphorylating and thereby inactivating the glycogen synthase [37,38]. GSK3α,β showed increased phosphorylation in control proliferating HepaFH3 and HepG2 cells compared to confluent HepaFH3 cells and PHHs, which is in line with the increased AKT phosphorylation in HepaFH3 cells and HepG2 cells (Figure 5D). Insulin induced P-GSK3α,β in PHH, although not significantly, in proliferating HepaFH3, in steatotic HepG2 cells but not in confluent HepaFH3 cells.

In line with the higher phosphorylation levels of AKT kinase, the downstream transcription factor and tumor suppressor Foxo1 also displayed higher phosphorylation in proliferating HepaFH3 and HepG2 cells, with phosphorylated Foxo1 being inactivated). Insulin induced P-Foxo1 only in PHHs and to a lower extent in proliferating HepaFH3 cells.

In addition to AKT kinases, ERK kinases are crucial for cell proliferation and survival and are activated by various extracellular signals, including growth factors, cytokines or mitogens [39]. HepG2 cells showed strongly elevated P-ERK levels in control samples compared to the other cell types, which is in line with their potentially increased proliferation (Figure 5E). Proliferating HepaFH3 cells showed no significant upregulation of P-ERK in control cells; however, a tendency was visible. Strikingly, P-ERK levels were even higher within steatosis for HepG2 cells and steatosis with insulin stimulus for both proliferating HepaFH3 cells and HepG2 cells, while PHHs and confluent HepaFH3 cells were not responsive to insulin regarding ERK activation.

Collectively, HepG2 and proliferating HepaFH3 cells displayed higher levels of P-AKT, P-GSK3α,β and P-Foxo1 compared to PHHs, while P-ERK was only elevated in HepG2 cells, which was even more pronounced in the steatotic state. PHHs and confluent HepaFH3 cells had strong similarities in signaling pathway activation levels in the unstimulated state. However, only PHHs were responsive to insulin stimulus. Besides P-ERK levels in HepG2 cells, steatosis did not significantly affect the analyzed signaling pathways.

## 4. Discussion

PHHs are the gold standard for toxicology and hepatology studies; however, their major limitations encompass their poor proliferation capacity and thereby reduced PHH cell numbers [40]. Upcyte hepatocytes were originally generated to overcome these limitations, and many studies have shown that upcyte hepatocytes maintain basic hepatocyte characteristics [25,41,42,43]. However, whether these cells are appropriate surrogates for PHHs regarding all cellular aspects remained elusive. We compared within our study the metabolic characteristics of upcyte hepatocytes HepaFH3 cells to PHHs and HepG2 cells and showed that HepaFH3 cells display closer similarities to HepG2 cells, suggesting HepaFH3 cells as a potential pre-malignant cellular model for HCC. The fact that an increasing amount of HCC arises directly from NAFLD indicates that a metabolic shift towards steatosis can foster tumorigenesis [6]. Therefore, we included steatosis in our study by using an established protocol [31]. We showed that HepaFH3 cells displayed morphological similarities to HepG2 cells and exhibited the same capacity of lipid storage as PHHs and HepG2 cells. Incubation of PHHs with fatty acids resulted in a strong increase in intracellular lipids, although this was not significant, probably due to the small study size of three donors. Steatosis induction resulted in some cell rounding and cell detachment of HepaFH3 cells and also slightly lower protein levels, but only mild lipotoxicity upon steatosis induction in the different cell types was observed by LDH and AST release, although not significantly and not by ALT release. Since LDH is the smallest protein of the three investigated markers, LDH is considered the most sensitive readout for viability assessment. As the release of ALT, the largest protein among these markers, was not observed, we concluded that the observed lipotoxic effect in our model is negligible.

The HepaFH3 cells showed many basic hepatocyte markers, including albumin, urea as well as AST and ALT synthesis, but to a lower extent than PHHs and partially to more similar levels to HepG2 cells. Tolosa et al. compared upcyte hepatocytes to normal hepatocytes and HepG2 cells and suggested that upcyte hepatocytes display similarities to normal hepatocytes with regard to basic hepatic metabolic functions, although albumin and urea levels in some upcyte hepatocyte clones were not as high as in normal hepatocytes [42]. Notably, we saw decreasing albumin levels with increasing passage numbers in HepaFH3 cells, which is an explanation for varying expression of hepatic markers. Confluent HepaFH3 cells displayed in some aspects similarities to PHHs, including higher AST and ALT levels as compared to proliferating HepaFH3 cells. Since confluent HepaFH3 cells additionally exerted the capacity of contact inhibition, confluent HepaFH3 cells could be used as a surrogate for PHHs for some metabolic assays where metabolism is not exhausted due to excessive proliferation. On the contrary, proliferating HepaFH3 cells and HepG2 cells displayed similarities in terms of increased glucose uptake and hence glycolytic activity, lactate production and metabolic activity. Increased lactate levels within tumor cells or strongly proliferating cells arise due to increased aerobic glycolysis, whereby pyruvate is converted into lactate, which cancer cells and proliferating cells use as a primary energy source (Warburg-effect) [12,44]. These data suggest that the Warburg effect is a general feature of energy shortage in fast proliferating hepatocytes. We detected even higher lactate levels in steatotic HepaFH3 cells. Since increased lactate in the circulation is known to reduce glucose uptake and induce insulin resistance [45,46], our data suggest that steatosis could predispose to insulin resistance, leading consequently to a fasting state.

Crucial signaling pathways in HCC pathogenesis include, among others, growth factor signaling via AKT and EGF kinases [47]; however, whether these signaling pathways are influenced by steatosis was not yet known. PI3K/AKT signaling is known to be activated in about 50% of HCC patients and was shown to block apoptosis in HCC cell lines and can therefore be considered a tumorigenic marker [48]. We show that both proliferating HepaFH3 cells and HepG2 cells displayed PI3K/AKT pathway activation, which was unaffected by steatosis, suggesting AKT activation as a marker for both proliferating and tumorigenic cells. Notably, proliferating diploid hepatocytes were shown to exert a higher oncogenic potential when subjected to tumor promoters [49]. The PI3K/AKT pathway is known to enhance glycolysis and lactate production and upregulates transporters for nutrient uptake [44]. This is in line with the observed metabolic characteristics of proliferating HepaFH3 cells and HepG2 cells. Higher phosphorylation of the AKT downstream signaling kinase GSK, known to inactivate glycogen synthase, was observed in proliferating HepaFH3 cells and HepG2 cells, which is in line with their reduced glycogen levels [37,38]. Another downstream signaling component of AKT, the tumor suppressor Foxo1, was shown to be more strongly phosphorylated in these cells, which is known to lead to its downregulation. This points as well toward increased proliferation in HepaFH3 cells and HepG2 cells since downregulation of Foxo1 was shown to drive proliferation in HCC cell lines [50]. Furthermore, as Foxo1 is a known regulator of gluconeogenesis [51], the phosphorylated and thereby reduced levels of Foxo1 suggest reduced gluconeogenesis.

ERK kinases are known to be activated in cancer [39]. Both HepG2 cells and proliferating HepaFH3 cells exerted elevated P-ERK levels, which were highest in HepG2 cells. This raises the possibility that extensive ERK activation might be a cancer-specific effect and a malignant signature in HCC. Steatosis and insulin stimulus resulted in even higher P-ERK levels in HepG2 cells, proposing that steatosis primes cancer cells for proliferation and survival. We induced steatosis in our study with a mixture of oleic and palmitic acid. Oleic acid has been shown to promote proliferation in HepG2 cells [52,53]. This would strongly suggest a tumorigenic effect of fatty acids in HCC pathogenesis. Our group showed in a previous study that high cellular loads of fatty acids induce ER stress in PHHs, resulting in reduced protein translation [54] and suggesting a concentration-dependent effect of fatty acids on cell proliferation. However, Giulitti et al. observed a blocking effect of oleic acid on the survival and proliferation of HCC cell lines Huh7 and Hep3B with reduced P ERK1/2 levels [55]. Possibly, the differences in our data compared to this study concerning ERK1/2 activation arise due to the fact that we used HepG2 cells and a mixture of oleic and palmitic acid. Oleic acid and palmitic acid have been shown to have opposite effects on HepG2 cell survival, with oleic acid being protective and palmitic acid being harmful [56]. Collectively, conflicting results exist as to whether fatty acids promote or inhibit cancer cell proliferation in HCCs, and further studies are required to understand this matter. However, in our study, extensive ERK activation in HepG2 cells might indicate a cancer-specific characteristic, which could discriminate proliferating and potentially regenerating cells from malignant cells in HCC.

The limitations of our study include the small study size of three PHH donors leading to difficulties in reaching statistical significance for the induction of steatosis. Furthermore, donor variance, including sex, age, pre-existing conditions and medication, could affect our results and have not been considered in this study. For our investigations, it would have been advantageous to use several upcyte hepatocyte clones or to compare the metabolic characteristics of HepaFH3 cells to further hepatoma cell lines such as Huh7 cells in addition to HepG2 cells. It remains to be answered whether the proliferative HepaFH3 cells can be discriminated from malignant cells by a proliferation signature, which would be useful for investigating HCC pathomechanisms. Our study revealed that extensive ERK kinase activation could be used for discriminating proliferating cells from malignant cells. A more in-depth analysis of HepaFH3 cells compared to HCC cell lines, primary hepatoma cells, and PHHs derived from HCCs or non-HCCs with a wide range of metabolic and differentiation markers is needed to differentiate the cells from each other, which will be part of future studies.

We strongly suggest the HepaFH3 cell line as a promising tool for analyzing HCC development mechanisms. Immortalized hepatic cell lines have been generated in previous studies from human hepatocytes, e.g., by the transduction of oncogenes including TERT, cyclin D1 or simian virus 40 large t (SV40 lt) antigen, and are a promising tool for hepatology studies [57,58]. Furthermore, hepatic cell lines with hepatoma cell origins, especially the HepaRG cell line, are a widely used tool in hepatology studies [59]. HepaRG cells have been generated from a hepatocellular carcinoma tumor from a single patient suffering from a hepatitis C infection and allow for infection with hepatitis B virus, which makes HepaRG cells a useful tool for studying viral hepatitis [59]. Furthermore, HepaRG cells can be differentiated into biliary or hepatocyte origin, making these cells a useful tool for studying hepatic differentiation [60]. The disadvantage of HepaRG cells is that they derive from one donor and that they can lose some hepatic markers due to malignant transformation [61]. The hepatic differentiation can be regained; however, this is time-consuming. The great advantage of the upcyte hepatocytes is that further cell lines from different donors can be generated, as can further cell lines from individuals with specific genetic variants or polymorphisms. It would be of interest to generate cell lines from individuals with the PNPLA3 polymorphism, which can predispose them to fatty liver disease [62]. Collectively, HepaFH3 cells, generated from hepatocytes of different donors of interest, constitute a useful system for analyzing pre-malignant mechanisms in hepatocytes during HCC development underlying steatosis.

## 5. Conclusions

Taken together, some similarities exist between confluent HepaFH3 cells and PHHs, but since confluent HepaFH3 cells lack the expression of specific hepatic markers, their usage as a PHH replacing model is restricted. However, their ability of contact inhibition would be of advantage for energy metabolism studies, including the steatosis in vitro model system. Proliferating HepaFH3 cells and HepG2 cells showed strong similarities in their metabolic characteristics; hence, we suggest proliferating HepaFH3 cells as a pre-malignant cellular model for HCC rather than as aa model for PHHs. The use of pre-oncologic and oncologic cellular liver systems for metabolic characterization of patient disease stages could help to better apply and optimize treatment for HCC patients and to predict which patients might progress to more advanced tumor stages.

## Figures and Tables

**Figure 1 biology-11-01195-f001:**
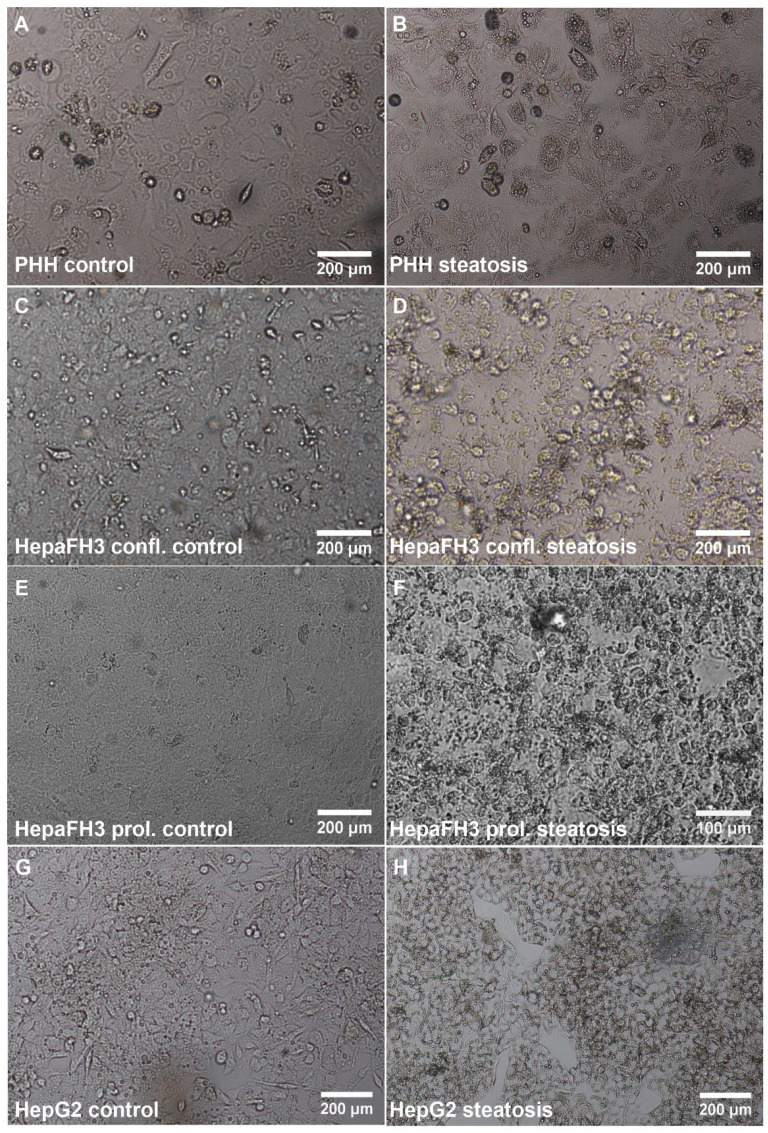
HepaFH3 cells exhibit a similar morphology to HepG2 cells. Light microscopy images of PHHs (**A**,**B**), confluent HepaFH3 cells (**C**,**D**), proliferating HepaFH3 cells (**E**,**F**) and HepG2 cells (**G**,**H**) in the control state and steatotic state are presented, respectively. Confl. = confluent, Prol. = proliferating.

**Figure 2 biology-11-01195-f002:**
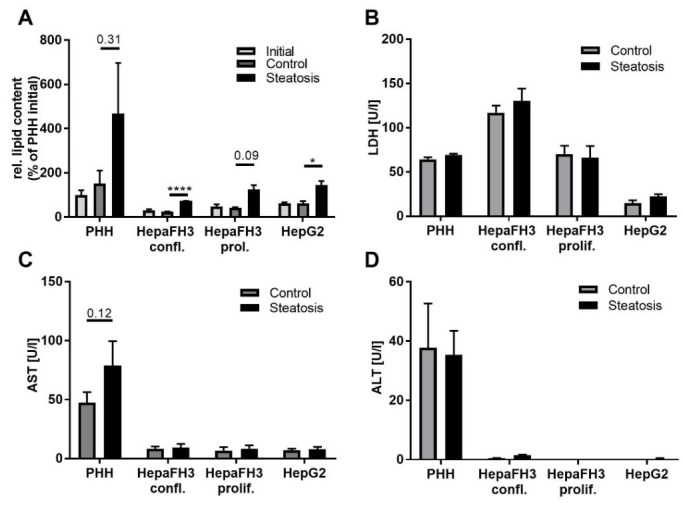
PHHs, HepaFH3 cells and HepG2 cells show the same ability of lipid storage, while lipid storage does not induce cytotoxicity. (**A**) Relative intracellular triglyceride accumulation, quantified by Oil Red O staining, is shown. PHHs, HepaFH3 confl., HepaFH3 prol. and HepG2 cells were incubated in control medium with or without oleate-palmitate mixture (2:1 ratio with a final concentration of 1 mM) for 24 h in order to induce steatosis. The initial lipid level was measured immediately before steatosis induction. Data were normalized to surface protein by SRB protein staining and are displayed as mean ± SEM from three independent experiments for each cell type. (**B**) Levels of LDH, (**C**) AST, and (**D**) ALT were analyzed in the cell supernatant to assess potential lipotoxicity. Data are displayed as mean + SEM from three independent experiments for each cell type. Significant differences (* *p* < 0.05; ** *p* < 0.01; *** *p* < 0.001; **** *p* < 0.0001) in relation to respective PHH samples is indicated. Significant differences within cell types are indicated by bars. Confl. = confluent, prol. = proliferating.

**Figure 3 biology-11-01195-f003:**
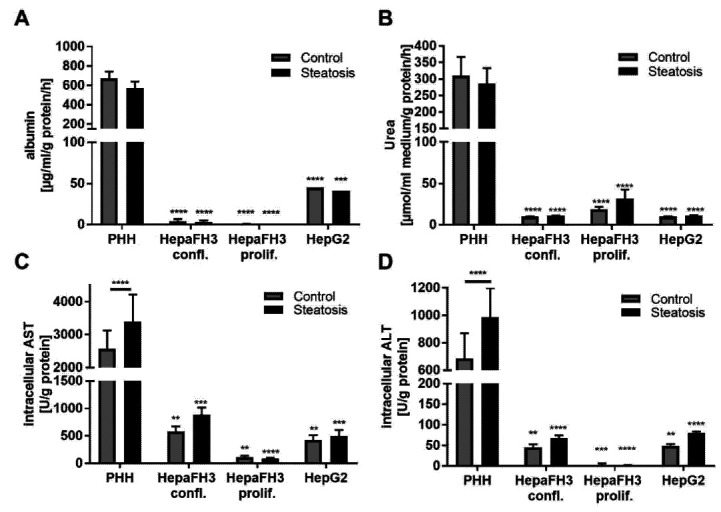
PHHs, but not HepaFH3 cells and HepG2 cells exhibit normal hepatic functions in terms of albumin and urea synthesis as well as intracellular AST and ALT levels. (**A**) Albumin level and (**B**) urea level in the cell supernatant between the different cell types with and without steatosis are given. Results were normalized to medium volume, total cellular protein by BCA assay and the period of time. (**C**) Intracellular AST levels and (**D**) intracellular ALT levels of the different cell types with and without steatosis are presented. The results were normalized to total cellular protein measured by BCA assay. Data are displayed as mean + SEM from three independent experiments for each cell type. Significant differences (* *p* < 0.05; ** *p* < 0.01; *** *p* < 0.001; **** *p* < 0.0001) in relation to respective PHH samples are indicated. Significant differences within cell types are indicated by bars. Confl. = confluent, prol. = proliferating.

**Figure 4 biology-11-01195-f004:**
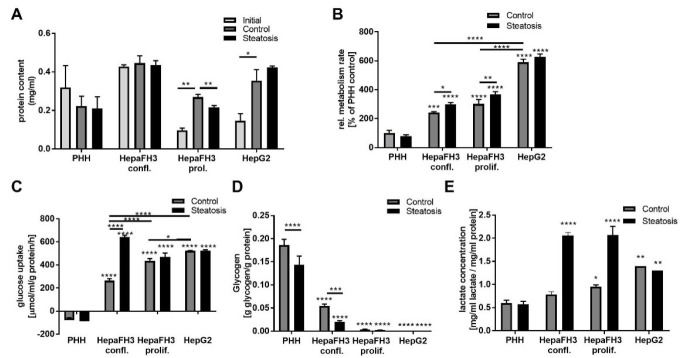
HepG2 and proliferative HepaFH3 cells show potentially increased proliferation, metabolic activity and glucose metabolism compared to PHHs. (**A**) Protein content as measured by BCA assay in cells 12 h after seeding (initial) and after further 24 h cell culture with (steatosis) or without steatosis (control) is given. (**B**) Relative metabolic activity measured by MTT assay in cells with and without steatosis is presented. (**C**) Glucose uptake was measured by determining glucose content of cell medium after 24 h cell culture from cells with and without steatosis. Glucose was measured using a glucose assay and was normalized to the medium volume, the total protein by BCA assay and the period of time. (**D**) Intracellular glycogen and lactate (**E**) were measured in cells with or without steatosis and were normalized to total protein content as measured by BCA assay. Data are displayed as mean + SEM from three independent experiments for each cell type. Significant differences (* *p* < 0.05; ** *p* < 0.01; *** *p* < 0.001; **** *p* < 0.0001) in relation to respective PHH samples are indicated. Significant differences within cell types are indicated by bars. Confl. = confluent, prol. = proliferating.

**Figure 5 biology-11-01195-f005:**
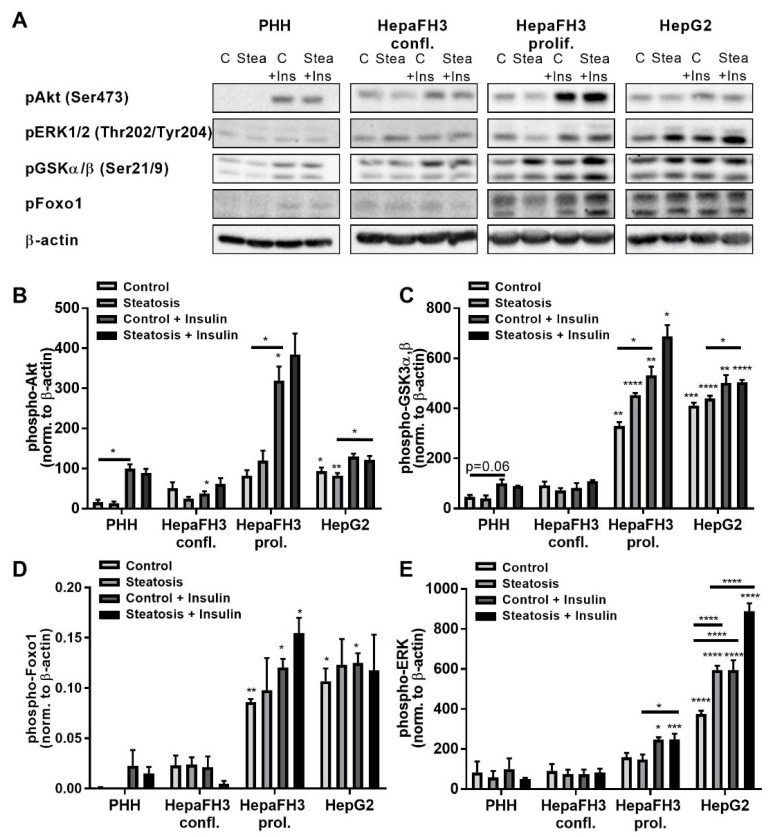
Increased AKT, ERK and Foxo1 phosphorylation in HepaFH3 and HepG2 cells suggests increased survival and proliferation compared to PHHs. (**A**) Western blot analysis and quantification of (**B**) phospho Akt, (**C**) phospho GSKαβ, (**D**) phospho Foxo1 and (**E**) phospho ERK of cells without steatosis (control), with steatosis (steatosis), cells without steatosis with insulin stimulus (control + insulin) and cells with steatosis with insulin stimulus (steatosis + insulin). β-actin served as loading control. Data are displayed as mean + SEM from three independent experiments for each cell type. Significant differences (* *p* < 0.05; ** *p* < 0.01; *** *p* < 0.001; **** *p* < 0.0001) in relation to respective PHH samples are indicated. Significant differences within cell types are indicated by bars. Confl. = confluent, prol. = proliferating.

**Table 1 biology-11-01195-t001:** Donor table with patient characteristics.

Donor	Age	Sex (w/m/d)	Diagnosis	Secondary Diagnosis
Donor 1	32	m	Brain metastasis	n.k.
Donor 2	49	w	Small bowel tumor with liver metastasis	n.k.
Donor 3	60	w	Klatskin tumor	hyperthyroidism

n.k. = not known.

## Data Availability

Not applicable.

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
