# Peer review of "Primary-like Human Hepatocytes Genetically Engineered to Obtain Proliferation Competence as a Capable Application for Energy Metabolism Experiments in In Vitro Oncologic Liver Models"

_biology, 2022, doi:10.3390/biology11081195_

Round 1

Reviewer 1 Report

The manuscript was well prepared. Moreover, the cited and their discussion are presented in good style. However, authors have some minor points that needs to be addressed.

I felt that HepaFH3 cells would be a useful premalignant cell model for HCC studies. I also felt that I would like to use them in my research on hepatic lipid metabolism in pre-cancerous states and early carcinogenesis.
 I would like to ask you if you have examined the expression of cholesterol synthesis and CYP enzymes in these cells, and if you plan to do so in the future.
If you are comparing the metabolism of different cells, please add this to this manuscript or consider it as a new manuscript in the future, as it is the main function of the liver.

minor points

・Please correct the type of font in Legends of Figure 2.

・Please unify the notation of units. 
   L204 µL   vs   L212 µl and more

・You must delete space in front of the percentage in Section 2.7

・L417, 432, 434, and more Fox01 -> Foxo1

Reviewer 2 Report

The authors generated a new human hepatic cell line HepaFH3, and characterized the hepatic features by comparing the cells to primary human hepatocytes and hepatoma cell line HepG2 in terms of morphology, intracellular lipid content, metabolic activity, secreted hepatocyte markers, and PI3K/AKT signaling pathway in response to steatosis induction.

Major concerns

1. Could the authors present microscopic images of PHH, HepaFH3, and HepG2 with higher resolution? The images in Fig.1 were too elusive to see more details, especially the morphology of PHH appeared to be different from normal primary human hepatocytes, which are characterized by cubic binucleate cell shape and clear refractive borders between hepatocytes.

2. For the experiments throughout the manuscript about the comparisons between HepaFH3, PHH, and HepG2 were lacking the details about the experimental setup, for example, what media were used for HepaFH3, PHH, and HepG2 for comparisons? The same medium or different medium. Different cell lines under different culture conditions were hard to compare with each other. How many cells were seeded in culture vessels? As shown in Fig.4A, the initial protein content for each cell differed a lot, so-called HepaFH3 diff. had the highest protein content, while HepaFH3 or HepG2 had the lowest. The differential initial protein content may be the reason why the four different cells had different protein content kinetics. Please use the same or similar level of initial protein content for the comparison.

3. In Fig.2, why was only intracellular lipid content normalized for quantification, but not other parameters like cLDH, cAST, or cALT? And what does the “c” ahead of the parameters mean?

4. Regarding the Western blot images in Fig.5, no total Akt, ERK, GSK FOXO1 were included, it’s not typical to only show the phosphorylated blots. Please include the blot results requested and compare the protein expression by normalizing the phosphorylated to total proteins. In addition, from the blot images in Fig.5, the audience cannot tell if the blots were exposed with the same time. Without the same exposure time, the comparisons would not be convincing.

5. The authors may consider not using the term “differentiated” HepaFH3, because there is not enough evidence to show HepaFH3 after 7 days of culture became differentiated status. Please refer to literature about HepaRG for whether or how to claim a cell line is well differentiated. In addition, a lot of immortalized hepatic cell lines have been generated and reported, HepaRG is the best known one. The authors should introduce the advance in hepatocyte immortalization, and discuss the rationale for why you generated a new hepatic cell line, what the pros and cons of the new cell line compared to others especially HepaRG.

Minor issues:

1. line 305 on page 11, “whixh” was wrongly spelled.

2. “PHH’s” should be PHHs.

3. Fig.5A, beta Actin but not Aktin.
